# An Inverted Perovskite Solar Cell with Good Comprehensive Performance Realized by Reducing the Concentration of Precursors

**DOI:** 10.3390/nano12101736

**Published:** 2022-05-19

**Authors:** Lijia Chen, Cunyun Xu, Yan Qin, Xiaofeng He, Hongyu Bian, Gaobo Xu, Lianbin Niu, Qunliang Song

**Affiliations:** 1College of Physics and Electronic Engineering, Chongqing Normal University, Chongqing 401331, China; niulb03@126.com; 2Institute for Clean Energy and Advanced Materials, School of Materials and Energy, Southwest University, Chongqing 400715, China; cyxuamos@foxmail.com (C.X.); hexf1992@163.com (X.H.); samuelhbian@163.com (H.B.); gbxu1805@163.com (G.X.); 3Foundation Department, Army Logistics Academy, Chongqing 401331, China; qy.yfds@foxmail.com

**Keywords:** inverted PSCs, comprehensive performance, material consumption, commercial application

## Abstract

Inverted perovskite solar cells (PSCs) exhibit great potential for industrial application thanks to their low complexity and low fabrication temperature. Aiming at commercial applications, it is necessary to comprehensively consider the material consumption and its corresponding electrical performance. Here, a simple strategy has been proposed to obtain inverted PSCs with comprehensive performance, that is, reaching an acceptable electrical performance by reducing the usage of perovskite. More precisely, the inverted PSCs, whose perovskite film is prepared by 1.0 M precursor, yields a power conversion efficiency (*PCE*) of 15.50%, fulfilling the requirement for real commercial application. In addition, the thickness of the electron transport layer (C_60_ in this work) in the above inverted PSCs was further optimized by comparing the simulated absorption spectrum, *J*-*V* characteristics and impedance with three different thicknesses of C_60_ layer. More excitingly, the optimized device displays high storage stability which maintains more than 90% of its initial *PCE* for 28 days. Therefore, our work provides a simple and cost-effective method to reach good comprehensive performance of inverted PSCs for commercial applications.

## 1. Introduction

Organic-inorganic hybrid perovskite materials, combining the facile processing of organic semiconductors with the high charge carrier transport property of inorganic materials, are one of the most promising candidates for the next generation of solar cells [1,2,3,4]. During the last decade, we have witnessed a dramatic increase in the power conversion efficiency (*PCE*) of perovskite solar cells (PSCs) from 3.8% to 25.7%, which is comparable to or even exceeds silicon-based solar cells [5,6]. It is worth pointing out that most of the highly efficient PSCs are achieved in regular (*n*-*i*-*p*) structures [7,8], in which *n* and *p* represent the electron transport layer (ETL) and hole transport layer (HTL), respectively [9,10]. As compared to regular devices, inverted PSCs (*p*-*i*-*n*) always exhibit a bit lower *PCE* but high opportunity for industrial application thanks to their relatively simple structure, with less energy consumption during fabrication and negligible hysteresis of device performance [11,12,13]. Therefore, much effort has been devoted to enhancing the *PCE* of inverted PSCs. For instance, thickness optimizations and electrical/optical characterizations have been employed in PSCs. Du et al. reported fabrication of relatively thick perovskite layers can considerably improve the average *PCE* under normal operational conditions [14]. Santos et al. reported optimizing film thicknesses in the search of major efficiency improvements, and the thickness of 400 nm for perovskite layer showed a benefit of significant absorption of light and reducing defects, which is facilitated by increasing quantum efficiency [15]. Koç et al. reported thickness optimizations of transport layers, perovskite and transparent conductive oxides to improve antireflection and light trapping properties, and to therefore maximize the photocurrent of PSCs [16]. Xu et al. reported high *PCE* inverted PSCs (reached 19.49%) based on MAPbI_3_(Cl) by employing self-woven deposited poly(3,4-ethylenedioxythiophene):polystyrenesulfonate(PEDOT:PSS) as HTL [11]. At the same time, Li et al. reported an inverted PSC based on a (FAPbI_3_)_0.95_(MAPbBr_3_)_0.05_ active layer with the highest *PCE* so far (>24%) by utilizing surface sulfidation treatment on the perovskite film [7].

Aside from the continuous enhancement of electrical performance in inverted PSCs, comprehensive consideration of material consumption and electrical device performance is another primary factor for their future development, especially for their commercial application. Unfortunately, there have been less reports on this issue. To lower material consumption, reducing the use of perovskite is likely the first method that comes to mind. However, the reduction of perovskite usage (e.g., reducing perovskite precursor concentration) directly leads to a decrease in the thickness of perovskite film, which is detrimental to light absorption and then inevitably decreases the *PCE* of the device [16,17]. Nevertheless, an appropriate reduction of *PCE* is acceptable if we consider that solar cells can reach a viable level for commercialization once their efficiency reaches ~15% [18]. Within this context, it is necessary to reach a balance between electrical performance and material consumption.

In this work, by tuning the concentration of perovskite precursor and comparing their corresponding electrical performance, a balance between material consumption and electrical performance has been achieved in inverted PSCs with the structure of ITO/PEDOT:PSS/MAPbI_3_(Cl)/C_60_/2,9-dimethyl-4,7-diphenyl-1,10-phenanthroline(BCP)/Ag. The inverted PSCs, whose perovskite film was prepared by 1 M perovskite precursor with a thickness of ~260 nm, exhibit better comprehensive performance for saving perovskite material and reaching a relatively high *PCE* of 15.5% at the same time. Additionally, the thickness of the C_60_ ETL behind the relatively thin perovskite film is further optimized to be 20 nm by comparing the simulated absorptivity of perovskite, electrical performance and impedance results with different thicknesses of C_60_ ETL. Furthermore, the optimized device shows relatively high storage stability whose *PCE* maintains more than 90% of its initial efficiency after 28 days without being encapsulated in the glove box. Taking all of this into account, our results provide a simple and cost-effective method for commercializing perovskite solar cells.

## 2. Experiments

### 2.1. Materials

Chemicals purchased from Advanced Election Technology Co., Ltd. in shenyang city, Liaoning province of China: methylammonium iodide (CH_3_NH_3_I, >99.99%), lead (II) iodide (PbI_2_, >99.99%), lead (II) chloride (PbCl_2_, >99.99%), PEDOT:PSS (1.3–1.7% in H_2_O dispersion). C_60_, BCP(>99%) and Ag were purchased from Xi’an Polymer Light Technology Corp (Xi’an, China). The solvents, including *N*,*N*-dimethylformide (DMF), chlorobenzene (CB) and dimethyl sulfoxide (DMSO), were brought from Sigma-Aldrich in Shanghai of China. PEDOT:PSS was mixed with deionized water (vol. ratio = 1:5). All the above materials were used as received without any further purification.

### 2.2. Sample Preparation

The origin perovskite precursor used in this work was prepared according to our previous report by dissolving 1.4 M CH_3_NH_3_I, 1.26 M PbI_2_, 0.14 mM PbCl_2_ in a co-solvent of DMSO and DMF (vol. ratio = 9:1) in a glove box followed by stirring overnight at room temperature [19]. The origin precursor was then diluted as required with the above co-solvent. To prepare the sample, the ITO substrates were firstly washed by an ultrasonic cleaning machine (KQ3200DV) with a mixture of a detergent (Decon 90) and deionized water at a volume ratio of 4% at 60 °C, and then repeatedly washed with deionized water at least three times. Subsequently, PEDOT:PSS (30 μL) mixed with deionized water was spin-coated onto the cleaned ITO substrates at 6000 rpm for 30 s in air, followed by heating on a hotplate at 120 °C for 20 min. A droplet of perovskite precursor (25 μL) with the target concentration was then spin-coated onto the pre-prepared PEDOT:PSS film at 400 rpm for 3 s and 4000 rpm for 30 s in the N_2_-filled glove box with oxygen and moisture levels < 3 ppm. Finally, a C_60_ film with different thicknesses (i.e., 20, 40 and 80 nm), a 6 nm BCP film and a 100 nm Ag electrode were deposited in sequence under high-vacuum (5 × 10^−5^ Pa) through shadow mask, resulting in an active area of 0.06 cm^2^.

### 2.3. Characterization

The current density–voltage (*J*-*V*) characteristics were recorded by Keithley 2400 source meter under 100 mW/cm^2^ simulated light (AM 1.5 G, Newport simulator: 94043A) in a glove box. During the test, the voltages were scanned from 1.2 to −0.2 V with a step of 20 mV and 0 ms delay time. The external quantum efficiency (EQE) was measured by a Newport QE system equipped with a lock-in amplifier in a glove box. The crystal and morphology characteristic of the perovskite absorption film were observed via X-ray diffraction system (Shimadzu XRD-7000) and scanning electron microscopy (SEM, JSM-6700F), respectively. The thicknesses of perovskite film prepared by different concentrations of precursor were measured by a profilometer (Kosaka ET 150). The reflection spectrum of the PSCs was carried out by UV–Visible spectrophotometer (UV-2550). The optical simulation was performed by an open source program based on the Python programming language. The detail of parameters, including refractive index, extinction coefficient and thickness of each layer, can be found in our previous work [20].

## 3. Results and Discussion

Figure 1a shows the schematic structure of an inverted PSC. As described in the introduction, our first preliminary attempt aims to obtain an inverted PSC with a better comprehensive performance, that is, achieving the balance between low material usage and relatively high PCE for commercial application. Therefore, a series of inverted PSCs, whose precursor concentrations are 1.4 M, 1.0 M and 0.6 M, respectively, were fabricated to measure and compare their electrical performance. It is worth pointing out that 1.4 M is the most commonly used precursor concentration in our group, therefore, it can be selected as reference in the following discussion. In addition, for the sake of simplicity, we will refer to *x* M device when the perovskite layer was prepared by another concentration, e.g., 1 M device represents the inverted PSCs whose perovskite film was prepared by 1 M precursor. As anticipated, the thicknesses of perovskite films gradually decrease with reducing concentrations of precursor since all precursors were spin-coated with the same coating parameters, as evidenced in Appendix A. Here, the thicknesses of perovskite films in the reference device, 1 M and 0.6 M device are~430, 265 and 190 nm, respectively.

Figure 1b displays the *J*-*V* characteristics of each inverted PSC prepared by different concentrations of perovskite precursors, and their corresponding photovoltaic parameters are extracted and summarized in Table 1. Clearly, decreasing the concentration of perovskite precursor leads to a decrease in device performance. Specifically, the reference device yields a *PCE* of 16.31% with a short-circuit current density (*J*_SC_) of 20.14 mA/cm^2^, an open-circuit voltage (*V*_OC_) of 1.025 V and fill factor (*FF*) of 79.1%, which is in good agreement with our previous results. ^2^ More excitingly, the 1 M device exhibits an acceptable electrical performance with a *PCE*, *J*_SC_, *V*_OC_ and *FF* of 15.50%, 18.53 mA/cm^2^, 1.03 V and 81.2%, respectively. Such laudable electrical performance indicates that the crystallinity and grain size of perovskite film prepared by 1 M precursor are not affected by decreasing its thickness, as evidenced in Appendix A, respectively. Moreover, a bit lower *PCE* is obtained in the 1 M device as compared to that of the reference device, which is mainly ascribed to the decrease in *J*_SC_ due to the uncomplete absorption of incident light caused by the relatively thin perovskite film. As the concentration of precursor is reduced further, the *PCE* of the 0.6 M device decreases to less than 14%, which cannot meet the requirements for commercial application. In addition, the above-mentioned change tendency was further confirmed by the external quantum efficiency (EQE) spectrum and their corresponding integral current (*J*_EQE_). As presented in Appendix A, a dramatically decreased EQE in the wavelength range from 580–750 nm is observed in both 1 M and 0.6 M devices as compared to that of the reference device. The calculated *J*_EQE_ of the reference device, 1 M device and 0.6 M device are 20.24 mA/cm^2^, 18.78 mA/cm^2^ and 18.32 mA/cm^2^, respectively, which is in accordance with the *J*_SC_ results extracted from *J*-*V* results. As shown in Appendix A, the perovskite layer decreased from 430 nm (1.4 M) to 265 nm (1.0 M), and the *PCE* decreased from 16.31% to 15.5%. That is to say, the material consumption of PSC devices is reduced by 38%, and the photoelectric conversion efficiency is reduced by only 5%. By considering the consumption of perovskite material and electrical performance from the perspective of commercial application, the 1 M device exhibits the best comprehensive performance, and our further investigations will be concentrated on this type of device.

As is well known, the thickness of the perovskite absorption layer is a critical parameter for determining the absorption of incident light and then the electrical performance of PSCs. According to previously published reports, it is necessary to point out that almost all incident light can be absorbed only when the thickness of perovskite layer exceeds 450 nm [21,22]. Therefore, the incident light cannot be completely absorbed by the perovskite layer in the 1 M device because its thickness is only~265 nm (Appendix A). In other terms, a small amount of incident light can reach to the electron transport layer (ETL) behind the perovskite layer and even the Ag electrode, leading to interference between the ETL and Ag. In such instances, it is reasonable to assume that the absorption of perovskite film in the 1 M device could be influenced by varying the thickness of the ETL, resulting in affecting its *J*-*V* characteristics.

Within this context, an optical simulation based on transmission matrix has been performed to get deeper insight into the influence of the thickness of the C_60_ ETL on light distribution in the 1 M device. This simulation was performed by an open source program written in the Python language [23], and all calculation parameters and details can be found in our previous paper [20]. During the simulation, 1 M devices with three different thicknesses (e.g., 20, 40 and 80 nm) of C_60_ ETL were selected to examine and compare. Figure 2a presents the simulated absorption of the perovskite layer in the 1 M device, which is significantly affected by tuning the thickness of the C_60_ ETL. More precisely, a constant absorption in the short wavelength range (400–550 nm) is observed in the above three devices, which is attributed to the very high absorption efficiency of perovskite in this wavelength range. In contrast, a gradual but obvious red-shift of the absorption peak together with absorption fluctuation in the long wavelength range (550–800 nm) are noticed when increasing the thickness of the C_60_ ETL. It should be highlighted that the long wavelength light (550–800 nm) is hardly absorbed by C_60_ material [23,24], thus, the absorption fluctuation originates from the interference of the unabsorbed light between the ETL and Ag electrode. As a consequence, the total absorption of perovskite film increases when increasing the thickness of the C_60_ ETL. This tendency was further confirmed by the measured absorption spectrum with the help of reflection spectrum (R(λ)) in the 1 M device (see Appendix A in Supporting Information). Using a simple calculation, 1 − R(λ), one can obtain the measured absorption spectrum. As displayed in Figure 2b, a similar absorption fluctuation in the long wavelength range is observed in the 1 M device with the above three thicknesses of C_60_ ETL. In sum, both the simulated and measured absorption spectrum imply that the *J*_SC_ of the 1 M device could be further increased because of the enhancement of total absorption after optimizing the thickness of the C_60_ ETL.

To confirm the above assumption, 1 M devices with 20 nm, 40 nm and 80 nm C_60_ ETLs (hereafter named as 1 M-20 nm, 1 M-40 nm and 1 M-80 nm device, respectively) were fabricated and measured to determine the optimized thickness of the C_60_ ETL. Contrary to our initial expectation, an almost constant *V*_OC_ but a moderate decrease of the *J*_SC_ is observed as the thickness of the C_60_ ETL increases, as shown in Figure 3a. All the photovoltaic parameters are summarized in Appendix A. The 1 M-40 nm device reaches a *PCE* of 15.11% with a *J*_SC_ of 18.25 mA/cm^2^, a *V*_OC_ of 1.03 V, and an *FF* of 80.4%. As the thickness of the C_60_ ETL increases to 80 nm, its efficiency is further decreased to 14.33%, and the *J*_SC_ is further reduced to 17.30 mA/cm^2^. Moreover, the gradual decrease of *J*_SC_ was further confirmed by EQE measurements and their corresponding integral current densities (see Appendix A in Supporting Information). It should be emphasized that the above results are confused, especially considering the increase of total absorption efficiency when increasing the thickness of the C_60_ ETL.

Aiming at a better understanding of the above puzzle, the electrochemical impedance spectrum (EIS) technique was carried out to analyze the internal resistance of inverted PSCs. Figure 3b shows the Nyquist plots of 1 M devices with the three different thicknesses of C_60_ ETL above, measured at a DC bias of *V*_OC_ and under 100 mW/cm^2^ illumination. The equivalent circuit is shown in the inset of Figure 3b, where a series resistance (*R*_s_) is ascribed to resistance of the ITO and wire electrode, the transportation resistance (*R*_ct_) is attributed to charge transfer, and the recombination resistance (*R*_rec_) is associated with the charge recombination rate. CPE is ideal capacitors and constant phase elements [25]. Additionally, an equivalent circuit has been employed to extract values related to internal resistance, including *R*_s_, *R*_ct_ and *R*_rec_. The fitted parameters are listed in Appendix A. First of all, the resistance difference of R_S_ is caused by the different positions of devices on ITO. Generally, the lower the *R*_s_ value, the higher the FF. Thus, a better FF is obtained in the 1 M-20 nm device. In addition, *R*_ct_ (and *R*_rec_) values extracted from 1 M-20 nm, 1 M-40 nm, and 1 M-80 nm devices are 337 (869), 352 (373) and 372(367) Ω, respectively. As is known, low *R*_tr_ is of benefit to charge carrier transport in the active layer while high *R*_rec_ is conducive to a reduction in charge recombination [26]. Therefore, benefiting from the lowest *R*_tr_ and largest *R*_rec_, the 1 M-20 nm device exhibits the highest *J*_SC_ as well as high *PCE*, although it has a relatively low total absorption efficient of its perovskite layer among the above three devices.

Additionally, the relationship between Voc and light intensity is employed to further explore charge recombination at the different thicknesses of C_60_ ETL in PSCs, as shown in Appendix A. It is known that the slope deviates from (kT/q), where k is the Boltzmann constant, T is temperature and q is the electric charge, reflecting trap-assisted recombination [27]. The PSC device based on 20 nm C_60_ exhibits a slope of 1.78 kT/q, indicating a lower trap-assisted recombination, while the slope with 40 nm C_60_ and 80 nm C_60_ are 1.80 kT/q and1.88 kT/q, respectively. These results further confirmed that the recombination has been suppressed with the C_60_ thickness decreasing, which is in agreement with the conclusion from Figure 3b.

Up to now, we have further optimized the thickness of the C_60_ ETL in the 1 M device, whose structure is ITO/PEDOT:PSS/perovskite (265 nm)/C_60_ (20 nm)/BCP (6 nm)/Ag (100 nm), to achieve the best comprehensive performance with a balance between electrical performance and material consumption. In regards to the stability of the 1 M-20 nm device, undoubtedly, stability is still one of biggest challenges for the commercialization of PSCs. Here, the storage stability measurement has been performed on the 1 M-20 nm device without encapsulation for more than 28 days in the glove box. For comparison, the reference device, whose perovskite film was prepared by 1.4 M precursor with 20 nm C_60_ ETL, was also tested in the same manner. As shown in Figure 4a, both devices undergo a slow and gradual degradation of the *PCE* values within the test period. After 28 days continuous recording, the *PCE* of the reference device maintains more than 90% of its initial value, which is further confirmed by the normalized results (see Appendix A). In particular, a similar *PCE* degradation behavior is noticed in the 1 M-20 nm device, suggesting the stability is not affected by decreasing the thickness of the perovskite layer covered by a 20 nm C_60_ ETL. The *PCE* degradation in both devices is mainly ascribed to the reduction of *J*_SC_ (see Figure 4b and Appendix A) because of almost no degradation of *V*_OC_ and FF (see Figure 4c,d and Appendix A) within the same period. All above stability results confirm good stability performance for the 1 M-20 nm device, which is of benefit to its further commercial application.

## 4. Conclusions

In summary, an inverted PSC with a better comprehensive performance has been achieved by simply reducing the concentration of perovskite precursor. The inverted PSCs, whose precursor concentration is 1.0 M, yield an acceptable *PCE* of 15.50% with 18.53 mA/cm^2^ of *J*_SC_, 1.03 V of *V*_OC_ and 81.2% of *FF*, reaching a balance between material consumption and its corresponding electrical performance for commercial application. Moreover, in the above 1 M device, its ETL thickness was further optimized (~20 nm) by comparing the simulated absorption spectrum, impedance spectrum and *J*-*V* characteristics with three different ETL thicknesses. Furthermore, the optimized device shows good storage stability, which maintains more than 90% of its initial value after one month stored in a glove box. Therefore, this study provides a simple and cost-effective method for reducing material usage in commercializing perovskite solar cells.

## Figures and Tables

**Figure 1 nanomaterials-12-01736-f001:**
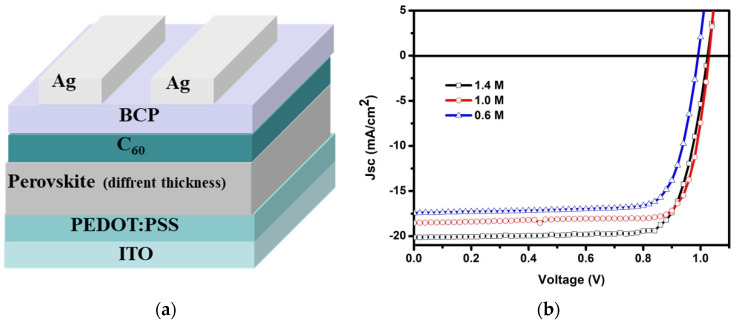
(**a**) Schematic architecture of the inverted PSCs. (**b**) *J*-*V* characterization of the inverted PSCs with perovskite layer prepared by different concentrations (1.4 M, 1.0 M and 0.6 M) of precursors.

**Figure 2 nanomaterials-12-01736-f002:**
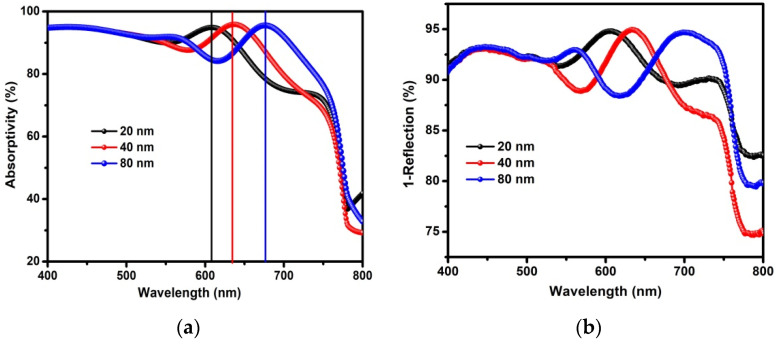
(**a**) The absorption of the 1 M device with different thickness (20 nm, 40 nm, 80 nm) of C_60_ ETL calculated by optical simulation. (**b**) Measured 1-reflection spectra for the 1 M device with different thickness of C_60_ ETL.

**Figure 3 nanomaterials-12-01736-f003:**
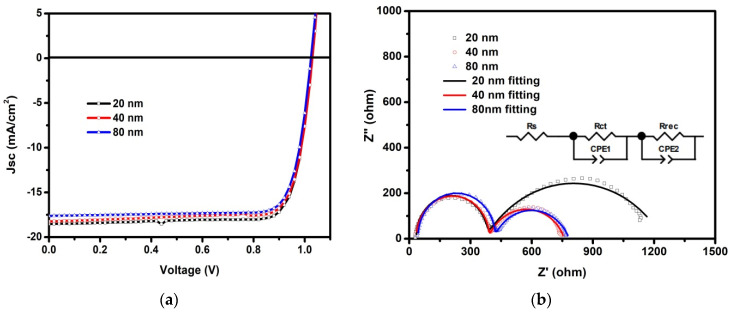
(**a**) *J*-*V* characterizations of the 1 M device with different thicknesses (20, 40, 80 nm) of C_60_ ETL. (**b**) Nyquist plots of 1 M devices with the above different thicknesses of C_60_ ETL. The inset is the equivalent circuit used for fitting.

**Figure 4 nanomaterials-12-01736-f004:**
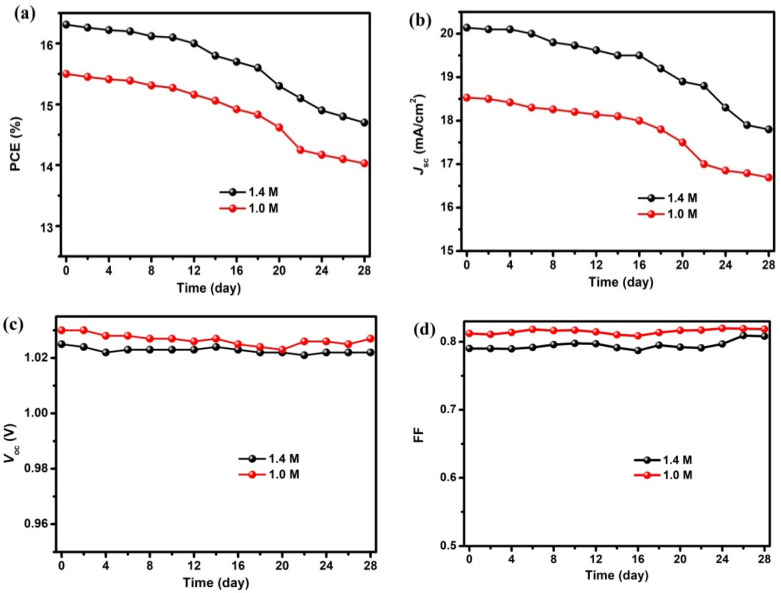
The stability of the 1 M-20 nm device and reference device stored in a glove box: (**a**) *PCE*, (**b**) *J*_SC_, (**c**) *V*_OC_ and (**d**) *FF*.

**Table 1 nanomaterials-12-01736-t001:** The photovoltaic parameters of inverted PSCs with the structure of ITO/PEDOT:PSS/perovskite (1.4 M, 1.0 M and 0.6 M)/C_60_ (20nm)/BCP (6 nm)/Ag.

Perovskite	*V*_OC_ (V)	*J*_SC_ (mA/cm^2^)	*FF* (%)	*PCE* (%)
1.4 M	1.025	20.14	79.1	16.31
1.0 M	1.030	18.53	81.2	15.50
0.6 M	0.991	17.43	78.7	13.59

## Data Availability

The data presented in this article is available on request from the corresponding author.

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
