# Peer review of "An Inverted Perovskite Solar Cell with Good Comprehensive Performance Realized by Reducing the Concentration of Precursors"

_nanomaterials, 2022, doi:10.3390/nano12101736_

Round 1

Reviewer 1 Report

The paper  is interesting, but the characterization by IES is not well referenced and even the way of the measurement is not the perfect one. Why Figure 3b has measured at 0,7V, should be measured at Voc, and 0.7V is less than the devices Voc. In the main text the equivalent circuit used is not referenced and it is not fully described, what is CPE-1 and CPE-2?

Why does the 20nm sample have a lower Rs? In addition, nowhere is the fitting of the equivalent circuit observed?

To compare the samples, they should measure the samples at different illuminations (using filters) and calculate n, which refers to the electronic recombination mechanism and see how the devices behave depending on the thickness of C60.

Reviewer 2 Report

In the manuscript entitled "An Inverted Perovskite Solar Cell with Good Comprehensive Performance Realized by Reducing the Concentration of Precursor", the authors demonstrated the thickness optimization of p-i-n inverted perovskite solar cells. The optimization was focused on two layers - perovskite and C60, and the devices were analyzed through the comparison with electrochemical impedance spectroscopy and UV-vis measurement.

However, it seems that the strategies that the authors demonstrated were more like typical optimization processes at laboratories, because they didn't provide any specific logics that can be thought to be novel or new. Thickness optimizations and electrical/optical characterizations based on them are not new. Besides, if the authors want to claim that the reduced thicknesses are helpful in not only reaching high-performance but also saving materials, then there should be a certain logic with sufficient evidence. Therefore, I don't recommend this article be published in Nanomaterials because of lacking novelty.

Round 2

Reviewer 1 Report

With the changes made. Now the article has improved a lot and is suitable for publication.

Reviewer 2 Report

Thank the authors for the response. The authors have revised the manuscript well along with my comments. Now I understand what they intend to mainly demonstrate in the work because of the explanation.